# A new snow module improves predictions of isotope-enabled MAIDENiso forest growth model

Ignacio Hermoso de Mendoza[1], Etienne Boucher[1], Fabio Gennaretti[2], Aliénor Lavergne[3], Robert Field[4], and Laia Andreu-Hayles[5,6,7]

[1]Centre de Recherche sur la dynamique du système Terre (GEOTOP) and Centre d'études nordiques, Université du Québec à Montréal (UQAM), Canada
[2]Institut de Recherche sur les Forêts (IRF), Université du Québec en Abitibi-Témiscamingue (UQAT), Canada
[3]Carbon Cycle Research Group, Space and Atmospheric Science, Physics Department, Imperial College London, United Kingdom
[4]NASA Goddard Institute for Space Studies, Dept. Applied Physics and Applied Mathematics, Columbia University, United States
[5]Tree-Ring Laboratory, Lamont-Doherty Earth Observatory of Columbia University, United States
[6]Ecological and Forestry Applications Research Centre (CREAF), Spain
[7]Catalan Institution for Research and Advanced Studies (ICREA), Spain

**Correspondence:** Ignacio Hermoso de Mendoza (ihmn.zgz@gmail.com)

**Abstract.** The representation of snow processes in forest growth models is necessary to accurately predict the hydrological cycle in boreal ecosystems and the isotopic signature of soil water extracted by trees, photosynthates and tree-ring cellulose. Yet, most process-based models do not include a snow module, consequently their simulations may be biased in cold environments. Here, we modified the MAIDENiso model to incorporate a new snow module that simulates snow accumulation, melting and sublimation, as well as thermal exchanges driving freezing and thawing of the snow and the soil. We tested these implementations in two sites in East and West Canada for black spruce (*Picea mariana* (Mill.) B.S.P.) and white spruce (*Picea glauca* (Moench) Voss) forests, respectively. The new snow module improves the skills of the model to predict components of the hydrological cycle. The MAIDENiso model is now able to reproduce the spring discharge peak and to simulate stable oxygen isotopes in tree-ring cellulose more realistically than in the original, snow-free version of the model. The new implementation also results in simulations with a higher contribution from the source water on the oxygen isotopic composition of the simulated cellulose, leading to more accurate estimates of cellulose isotopic composition. Future work may include the development of inverse modelling with this new version of MAIDENiso to produce robust reconstructions of the hydrological cycle and isotope processes in cold environments.

## 1 Introduction

In boreal regions of Canada and Alaska, snow represents about 30-50% of total precipitation (Mesinger et al., 2006). This feature has notable influence on hydrological and ecological system functioning in these cold environments (Beria et al., 2018). From a hydrological perspective, snowpack dynamics greatly influence water infiltration in soils, groundwater and aquifer replenishment, runoff production and water supplies to both natural and artificial water bodies during spring flood (Li et al.,

2017; Barnhart et al., 2016; Berghuijs et al., 2014). From an ecological perspective, snowpack accumulation protects exposed

plant tissues and organs against cold winds (Boivin and Bégin, 1997). Snow melt contributes to mitigate the negative impacts of droughts on tree growth (St. George et al., 2009), while affecting photosynthesis (Perkins and Swetnam, 1996; Peterson and Peterson, 1994). Snowpack dynamics also have the potential to alter heat fluxes, temperature and depth of freezing in soils, all of which can impact the timing of critical ecophysiological processes that drive growth in high latitude forest stands.

For decades, tree-ring proxies such as ring widths (Nicault et al., 2015; Ols et al., 2018), wood density (Boucher et al., 2017)

or stable isotope ratios of tree-ring cellulose (Naulier et al., 2015b, 2014; Porter et al., 2014) have been used to track inter-annual changes in forest response to climate variability. Most studies emphasized on the dominant role of summer temperatures on key ecophysiological processes controlling proxy formation. This has helped to clarify the response mechanisms of the boreal forest to growing season temperatures (Gennaretti et al., 2017a) and enabled long, millennial summer temperature reconstructions to be produced in this region (Gennaretti et al., 2017a; Naulier et al., 2015a; Gennaretti et al., 2017b). However, despite

their ecological and hydrological significance, snow-related processes were rarely taken into account in these tree-ring studies (Coulthard et al., 2021; Woodhouse, 2003; Huang et al., 2019; Zhu et al., 2021). Consequently, the impact of these changes in snow cover properties (Meredith et al., 2019) on vegetation growth and ecolophysiological response remain highly uncertain.

Predicting the effect of snow dynamics on tree growth is a complex task as both phenomena occur in distinct seasons (Coulthard et al., 2021). Inter-seasonal heat and moisture fluxes attributable to snow need to be accounted for in order to

accurately model the impact of snow on tree-ring formation. The timing and magnitude of these transfers, however, result from a complex interplay between snowpack properties (snow depth, density and water content) and processes that control snow accumulation and melt (precipitation, sublimation, redistribution by wind, rain on snow events, among others) (Rutter et al., 2009). These transfers also modify the isotopic signature of the water used by trees. Indeed, snow is more depleted in the lighter isotope $^{18}O$ than rainfall (Kurita et al., 2004), but sublimation-driven enrichment of snow may also change the isotopic

composition of the source water used by trees. Ultimately, this should be recorded in $\delta^{18}O$ of tree-ring cellulose (Beria et al., 2018). Correlation-based tree-ring analyses based on statistical relationships cannot take into account this mechanistic level of complexity, and thus there is a need to explicitly integrate snow dynamics in forest growth models.

Process-based models developed for simulating tree growth are important tools to study the relationship between climate and tree-ring proxies (Guiot et al., 2014). These models are driven by meteorological and environmental variables, and inte-

grate a wide number of equations that represent state of the art knowledge on how physical and ecophysiological processes determine tree response to climate variability. A number of process-based models have been developed over the years, such as the Vaganov-Shashkin (VS) model (Fritts et al., 1991; Vaganov et al., 2006; Shishov et al., 2016), MAIDEN (Misson, 2004), StandLeap (Girardin et al., 2008), CAMBIUM (Drew et al., 2010), ECOPHYS (Hölttä et al., 2010), Biome3 (Rathgeber et al., 2003) or the T-model (Li et al., 2014). Despite the importance of snow for tree growth, most process-based models do not

include a snow module, mostly because they were not designed to be used in boreal and alpine environments or even in mid-latitude temperate forests where snow accumulates during winter. Among the previously mentioned models, exceptions are the Vaganov-Shashkin (VS) model (Shishov et al., 2016) and the Biome3 model (Rathgeber et al., 2003), which incorporate basic models of snow accumulation and melt driven by air temperature, but do not consider processes such as sublimation, en-

ergy balance or stable isotope fractionation of water isotopes during the cold season. Among the available models, MAIDEN
(Misson, 2004) was specifically designed to improve the interpretation of tree-ring proxies based on our knowledge about eco-
physiological processes and relationships between climate and tree growth. MAIDEN simulates the water and carbon fluxes
exchanged between forests and the atmosphere, including the influence of phenology on the production and allocation of car-
bon to different parts of the tree. Because it requires a very limited number of meteorological inputs, the application of the
model is possible in regions where data is scarce. The isotope-enabled version MAIDENiso (Danis et al., 2012) incorporates
calculations of the stable isotopic composition of oxygen ($\delta^{18}$O) and carbon ($\delta^{13}$C) in the different components of the tree.
MAIDEN was originally created for tree species in Mediterranean climates, and has been optimized for *Quercus petraea*
(Matt.) Liebl. and 12 Mediterranean species (Misson, 2004; Gaucherel et al., 2008; Boucher et al., 2014; Gea-Izquierdo et al.,
2015). Since then, the phenology and physiological processes have been adapted to simulate tree radial growth in boreal north-
eastern American forests (Gennaretti et al., 2017a) and used to simulate tree-ring cellulose $\delta^{18}$O in boreal and temperate forests
of eastern Canada and southern South America (Lavergne et al., 2017). MAIDENiso provides two main advantages over other
process-based models: 1) The outputs are directly comparable to tree-ring proxies. 2) It is an isotope-enabled model, allow-
ing to track down the origin of the climate signal recorded therein. However, the use of MAIDENiso in high-latitude forests
has been limited by the fact that its hydrological cycle was never adapted to boreal conditions, and the lack of an adequate
representation of snow dynamics.

Here, we incorporate a new snow module in MAIDENiso and test the simulated data against real observations. This module
is driven by a new thermal conduction model to improve the simulations when the model is used in cold environments where
snow is present. This snow module allows MAIDENiso to reproduce the basic dynamics of the snowpack, targeting a more
realistic water balance and water isotope frationation sequence by representing the $\delta^{18}$O signal of snowfall, the sublimative
fractionation at the snow surface and its final imprint in tree-ring cellulose (TRC). Despite this added complexity on processes,
the snow model can work with the same small number of environmental variables that MAIDENiso currently requires. In this
study, we evaluate the impact of the new snow module on the simulation of soil moisture, water outflux and the $\delta^{18}$O signal
in soil and TRC in two forest sites in Canada: a black spruce (*Picea mariana* (Mill.) B.S.P.) forest in the Caniapiscau basin
(Québec) and a white spruce (*Picea glauca* (Moench) Voss) forest in Tungsten (Northwest Territories).

## 2 Materials and methods

### 2.1 MAIDENiso model

#### 2.1.1 Original model

MAIDENiso (Misson, 2004; Danis et al., 2012; Gea-Izquierdo et al., 2015; Gennaretti et al., 2017a) simulates the mechani-
cal and physiological processes of a tree and its immediate environment. The model requires daily meteorological inputs of
maximum and minimum temperature, precipitation, and atmospheric $CO_2$ concentration (optional inputs are relative humidity,
radiation, wind speed and atmospheric $\delta^{13}$C). MAIDENiso simulates gross primary production (GPP) and carbon allocation

on a daily basis based on inputs of meteorological and tree phenological data. Carbon is allocated explicitly to several pools (leaves, roots, stem and a carbon reservoir) using mechanistic rules dependent on phenology. A diagram of the model is shown in Fig. 1 with the original components of the model depicted in black. These original components include the photosynthesis module and the isotopic module, which are described in the sections A and B of the appendix, respectively.

MAIDENiso simulates the hydrological processes in the immediate environment around the tree: at canopy (interception and canopy evaporation), ground surface (infiltration, evaporation and runoff) and underground (hydraulic transfers and root absorption) levels. These processes are modelled through a series of water pools and fluxes (Fig. 2). For instance, the canopy can intercept a portion of the precipitation water up to a maximum determined by the leaf area index (LAI), which can be evaporated or dripped to the ground overnight. The surface of the soil cannot hold any stagnant water, so daily incoming water from throughfall infiltrates the soil (up to a maximum determined by soil properties) or exits the system as runoff. The soil consists of four layers of distinct thickness, with porosity and hydraulic conductivity determined by the composition of the soil, and water moves between these layers following Darcy's law. Soil water is replenished through infiltration and depleted by root absorption for transpiration (at all layers), soil evaporation (at the upper layer) and drainage (at the bottom layer).

The original version of MAIDENiso (Gennaretti et al., 2017a; Danis et al., 2012) includes one snow layer, where snow accumulates and melts following changes in atmospheric temperature. This module was implemented to simulate snow reflectivity and thus changes in albedo as part of the calculation of the energy budget. However, this was a side-calculation that did not interact with any of the other subsystems in MAIDENiso, and thus the accumulated and melted snow was not taken into account in the water balance calculation. In addition, all water pools and fluxes in the model were liquid regardless of temperature. In boreal climate, this previous version of MAIDENiso was thus unable to predict snow accumulation during winter, and therefore did not include different source water signatures due to snowfall instead of rainfall, the fractionation of $\delta^{18}$O due to snow sublimation or the rapid melting of snow in spring. Therefore, this previous version of the model simulated unrealistic soil moisture and hydrological outflux (drainage and runoff) and too depleted values of $\delta^{18}$O in source water in spring.

### 2.1.2 New implementations in the model

The hydrology in the new version of MAIDENiso incorporates several pools of solid water: a canopy snow pool, a single snow layer on top of the soil and a pool of ice in each soil layer (Fig. 1). In addition, the snow layer is able to hold liquid water in its porous space, adding a new pool of liquid water. These water pools and the new water fluxes are shown in Fig. 2.

Input precipitation to the system is first partitioned into rainfall and snowfall based on the average daily temperature, following a linear partition between -2 °C and 4 °C (McCabe and Wolock, 2009). Following the same interception rule as liquid water, snow can be intercepted by the canopy and added to the canopy snow pool with a maximum capacity determined by LAI, from where snow can sublimate. However, while the canopy water pool always becomes empty at the end of each day, the canopy snow pool does not. Canopy snow can still drip to the ground based on atmospheric temperature following a drip model taken from the Community Land Model version 5 (CLM5) (Lawrence et al., 2019).

A single, uniform snow layer can cover the uppermost soil layer fully or partially, keeping track of snow thickness and the masses of snow and liquid water in the layer and calculating the density of the layer dynamically. Freezing transfers water

to snow mass without changing thickness, thus increasing density (with pure ice density as maximum), while melting and sublimation remove snow mass but keep density constant. The snow layer is forced to have a minimum thickness of 0.1 m (which is needed for numerical convergence), so the partial snow cover is decreased to avoid a thickness below this minimum (i.e. partial cover of zero when no snow is present). Snowfall always accumulate over the existing snow layer, increasing mass and thickness according to a temperature-variable density model of newly fallen snow (van Kampenhout et al., 2017). In contrast, the portion of rainfall that hits the snow layer is determined by the partial snow cover. Sublimation from the snow layer is calculated by modifying the version of the Penman-Monteith equation (Stigter et al., 2018) as follows:

$$\lambda E_{pot,snow} = \frac{\Delta \cdot R + \rho_{air} \cdot C_P \cdot \delta_{atm}/r_a}{\Delta + \gamma} \tag{1}$$

$\Delta$ (kPa $^{\circ}$C$^{-1}$) is the gradient of the saturation vapor pressure curve, $\gamma$ (kPa $^{\circ}$C$^{-1}$) is the psychrometric constant (Loescher et al., 2009), $R$ (MJ) is the net radiation over the snow surface, $\rho_{air}$ (kg m$^{-1}$) is the air density, $C_P$ (MJ kg$^{-1}$) is the specific heat of dry air, and $r_a$ (s m$^{-1}$) is the aerodynamic resistance to water vapor transfer. $r_a$ typically depends on several factors, such as wind (Blanken and Black, 2004). However, because wind data is not usually available in tree-ring sites, some assumptions need to be done to use the equation above. Here, we assume a constant $r_a$ that is optimized for each site using the available data of snowfall and snow pile's thickness and mass.

A pool of ice has been added to each soil layer. The pools of liquid water and solid water (ice) in each layer compete for the same porous space, thus the ice content of a soil layer decreases its effective porosity. This decreases both the maximum amount of liquid water that a layer can hold and the hydraulic conductivity of the layer. Soil ice increases when soil temperature is below 0 $^{\circ}$C and decreases when soil temperature is above 0 $^{\circ}$C.

To calculate the change in water phase from solid to liquid in both the snow and soil layers, we have added a one-dimensional (vertical) thermal conduction model largely based on CLM5 (Lawrence et al., 2019). In this model, the system composed by the snow-soil layers is bounded at the top (as soil or snow) by the heat flux from the overlying atmosphere and at the bottom by a constant value representing the geothermal heat flux. The amount of water (or ice) that freezes (or melts) is calculated from the deficit (or excess) of energy to keep the temperature of the layer at 0 $^{\circ}$C.

The new implementation of snow in MAIDENiso is now able to reproduce the dynamics of the snowpack, which is connected to the rest of the components of the model (Fig. 1). The accumulation of winter precipitation increases the amount of water available in the soil in spring, which in turn may favor the onset of photosynthesis. A higher photosynthetic activity results in more carbon assimilated by the canopy, potentially leading to a shorter budburst phase. A diagram showing the links between the different components in MAIDENiso is shown in Fig. 1.

The new MAIDENiso version also includes new isotopic fractionation processes for the sublimative fluxes and for the phase changes between liquid water and ice. In cold regions where snowfall is a considerable portion of the yearly precipitation, fractionation from snow sublimation is expected to produce a significant enrichment of the $\delta^{18}$O isotopes in the snow layer, which after melting may be incorporated in the soil water, and ultimately reflected in TRC.

Given the already high number of parameters in MAIDENiso (121 in the new version with snow, 117 in the previous version), one of our goals during the development of the new snow module has been to keep the number of new free parameters to the minimum possible. Despite the complexity and the new processes incorporated to the model, the new snow module only added four new parameters to MAIDENiso, which are listed in Table 1. Three of them are site parameters (determined externally to MAIDENiso) that correspond to a linear regression model of precipitation $\delta^{18}O$ (more information in subsection 2.3). Therefore, we only added a single free parameter that requires calibration: the resistance to vapor transfer $r_a$. This parameter controls snow sublimation, which fundamentally depends on wind speed and therefore varies considerably between sites, requiring independent calibration at each site that as explained above we computed using observations of snowfall and snowpile's thickness and mass because wind data is not available.

## 2.2 Calibration of MAIDENiso

Different parameters that are species-dependent and site-dependent need to be defined before running MAIDENiso at a particular site. Most of these parameters can be obtained from direct observations at the studied site, such as the characteristics of the soil (composition and depth) or the root-leaf proportions of the tree species. When the values of the parameters are unknown, these are calibrated through a Bayesian optimization algorithm described in detail in Gennaretti et al. (2017a). This optimization is based on Markov chain Monte Carlo (MCMC) sampling that retains combinations (blocks) of parameters that satisfy a condition, maximizing the coincidence between a series of observations and the equivalent products simulated by MAIDENiso. Here, we used 50 independent chains of parameter blocks and selected the most optimal block of parameters (called the "plausible block").

The series of observations used in the MCMC consist of observed time series of snow (depth or mass of the snowpile), GPP and $\delta^{18}O$ in Tree-Ring Cellulose ($\delta^{18}O_{TRC}$). The parameters to be determined via MCMC for each component of the model are the following:

- Snowpile: 1 parameter, $r_a$ from Eq. (1). Calibrated by comparing observed and simulated daily Snow Depth (SNDP).

- GPP: 6 parameters (see appendix A). Calibrated by comparing observed and simulated daily GPP.

- $\delta^{18}O_{TRC}$: 3 parameters (see appendix B), $f_0$, $\epsilon_0$ and $\epsilon_k$ in Eqs. (B1) and (B2). Calibrated by comparing observed and simulated yearly $\delta^{18}O_{TRC}$.

Because the new and the original versions of MAIDENiso behave differently, an independent calibration of the parameters is needed to run each of them. Note that the original version of MAIDENiso does not need to be calibrated for the snow parameters because it does not include a snow module.

Some parameters can influence more than one process indirectly, e.g. the snowpile affects source water and therefore $\delta^{18}O_{TRC}$, or the GPP parameters control the amount of carbon produced, which in turn affects $\delta^{18}O_{TRC}$. To avoid that the calibration of some processes affects parameters that are already calibrated, the parameter sets need to be calibrated in a specific order: snow first, GPP second, and lastly $\delta^{18}O_{TRC}$.

## 2.3 Study sites and input meteorological data

The study tree-ring sites are located in Tungsten, Northwest Territories, Yukon border (61.98° N, 128.25° W, 1145 m a.s.l.) and in the Caniapiscau basin, Quebec (54.86° N, 69.72° W, 530 m a.s.l.).

MAIDENiso requires daily meteorological inputs for a continuous period of time overlapping with the period of available observations. Daily $CO_2$ data were obtained from the Mauna Loa Observatory observations (Keeling et al., 1976) corrected with the CarbonTracker measurement and modeling system (Peters et al., 2007).

The closest meteorological stations to the study sites were located 100 km away from Tungsten and 186 km from Caniapiscau. Therefore, temperature and precipitation data were taken from the NARR (North American Reanalysis) dataset (Mesinger et al., 2006) at the coordinates of the studied sites. The NARR has a spatial resolution of 32.5 km × 32.5 km and spans for the period 1979-2013. Meteorological inputs were also needed at two additional sites to calibrate GPP, which are described in detail in section 2.4. During the period 1979-2013 based on NARR, average summer temperatures (June-July-August) ranged 5.4-12.6 °C in Tungsten and 8.2-16.2 °C in Caniapiscau. Yearly temperatures at Tungsten are stable during the whole period, while at Caniapiscau average temperatures steadily rise after 1990 by 0.1 °C per year. Average yearly precipitation during this period were 584 mm and 796 mm in Tungsten and Caniapiscau, respectively. The 56% of the yearly precipitations in Tungsten was snowfall, while in the warmer site of Caniapiscau snow was only 45%.

MAIDENiso also needs information about $\delta^{18}O_P$. Two different approaches can be used to infer $\delta^{18}O_P$. The first and most direct way is to use daily values of $\delta^{18}O_P$ as another meteorological input. However, these values are often not available. The second approach, which we used here, is to use precipitation (P, mm), air temperature ($T_{air}$, °C) and $\delta^{18}O_P$ from an observed dataset to obtain a linear regression model for daily values of $\delta^{18}O_P$ based on temperature and precipitation:

$$\delta^{18}O_P = a \cdot T_{air} + b \cdot P + c \tag{2}$$

This approach has the advantage that, once the model is obtained, the parameters can be used with a different dataset of air temperature and precipitation to obtain the corresponding $\delta^{18}O_P$ values. In this paper, we calibrated this regression model using meteorological data from the gridded dataset IsoGSM (Yoshimura et al., 2008) from the grid points that contain the coordinates of the Tungsten and Caniapiscau sites. We discarded the direct use of the IsoGSM meteorological and $\delta^{18}O_P$ data into MAIDENiso because the precipitation amounts derived from the IsoGSM were too low compared to the amounts from observations recorded in meteorological stations nearby the study sites. However, the IsoGSM meteorological data was still useful to obtain the parameters for our regression model. We obtain different equations for liquid (rainfall: $a_{rain}$, $b_{rain}$, $c_{rain}$) and solid precipitation (snowfall: $a_{snow}$, $b_{snow}$, $c_{snow}$) using separately data corresponding to temperatures below -4 °C for snowfall and higher than 2 °C for rainfall (see Table 2).

## 2.4 Tree-Ring $\delta^{18}O$, GPP and snow data

We used published $\delta^{18}O_{TRC}$ chronologies for Tungsten (Field et al., 2021) and Caniapiscau (Nicault et al., 2014). These chronologies span between 1900-2003 for Tungsten and 1948-2013 for Caniapiscau, however for this study we used the isotopic records for periods that overlap with the NARR meteorology: 1979-2003 for Tungsten and 1979-2013 for Caniapiscau.

We used GPP data available from the closest eddy covariance flux stations to estimate the parameters controlling GPP, assuming that the obtained parameters were similar at the studied sites. To calibrate GPP in Tungsten, we used the University of Alaska Fairbanks (Uaf) station from the Ameriflux network (64.87° N, 147.85° W; data period 2003-2018;Ueyama et al. (2021)) at 1023 km from our study site. For Caniapiscau, we obtained daily GPP data from an eddy covariance station located in a mature black spruce forest in northern Quebec ("Quebec Eastern Old Black Spruce station" - EOBS; 49.69° N, 74.34° W; http://fluxnet.ornl.gov/site/269; data period 2003-2010, Bergeron et al. (2007)) at 650 km from our study site. Although these eddy covariance flux stations are geographically distant from our study sites, they provide GPP data for the same tree species in our sites. Because the parameters used to calibrate GPP are more related to species-specific traits (Gennaretti et al., 2017a) than environmental conditions at a given site, it is a reasonable assumption to calibrate GPP at these stations and use the obtained GPP parameters in our study sites. MAIDENiso simulated GPP at both stations, using the following meteorological inputs. For the Uaf station, we used the meteorological inputs available at the station. For the EOBS site, the meteorological inputs were taken from the gridded interpolated Canadian database of daily minimum-maximum temperature and precipitation for 1950-2015 (Hutchinson et al., 2009), used in Gennaretti et al. (2017a).

In-situ snowpile data are needed to test the predictive skills of MAIDENiso to simulate the snowpile. The Snow Water Equivalent (SWE) is the ideal snowpile data to use because addition (from precipitation) and removal (from sublimation and melting) of snow to or from the snowpile is calculated in units of mass. Alternatively, snow depth (SNDP) data, most commonly available, can be used as well to compare with observations but requires knowledge about snow density. SWE field measurements were only available for the Caniapiscau site at discrete (biweekly) intervals during winter and early spring between 1971-1993 (data provided by Hydro-Québec, personal communication). Therefore, in order to make the results from both sites comparable, we used SNDP data to calibrate the snowpile at Tungsten and Caniapiscau and only used the SWE measurements at Caniapiscau to validate the simulations. The SNDP data (1979-2013) were extracted from NARR for Caniapicau and from observations of a meteorological station for Tungsten.

## 2.5 Model evaluation and experiments

To evaluate the agreement between observed and simulated $\delta^{18}O_{TRC}$ for the two versions of MAIDENiso, we calculated the Pearson correlation coefficient and associated p-values ($p < 0.05$ were considered significant). To determine that the simulated $\delta^{18}O$ at the leaf and cellulose level were statistically different, we used the Welch t-test (which tests the null hypothesis that the difference between the means of two curves is zero).

While MAIDENiso does not calculate river discharge as an output (which would be possible through the implementation of a routing model), water discharge (water leaving the system in liquid form) can be calculated as the addition of runoff (water overflowing the infiltration capacity of the soil) and drainage (downwards water flux from the lowest soil layer), and be compared to measurements of river discharge. For this study these comparisons were done for only for Caniaspiscau due to the avaibility of river discharge observations. To evaluate the coincidence between the observed and simulated water discharge, we used the Nash-Sutcliffe model Efficiency (NSE) coefficient (Nash and Sutcliffe, 1970), which is equivalent to a coefficient of

determination:

$$\text{NSE} = 1 - \frac{\sum_t (Q_{sim}^t - Q_{obs}^t)^2}{\sum_t (Q_{obs}^t - \overline{Q}_{obs})^2} \tag{3}$$

where $Q_{sim}^t$ and $Q_{obs}^t$ are the simulated and observed discharge at time t, respectively. The NSE ranges between $-\infty$ and $+1$. To facilitate the interpretation of NSE, we rescaled the NSE within the range of (0,1) with the Normalized Nash-Sutcliffe Efficiency (NNSE) coefficient (Nossent and Bauwens, 2011):

$$\text{NNSE} = \frac{1}{2 - \text{NSE}} \tag{4}$$

Another useful metric is the Kling-Gupta efficiency (KGE) (Gupta et al., 2009), which addresses several shortcomings of the NSE and is increasingly used for model calibration and evaluation:

$$\text{KGE} = 1 - \sqrt{(r-1)^2 - (\frac{\sigma_{sim}}{\sigma_{obs}} - 1)^2 + (\frac{\mu_{sim}}{\mu_{obs}} - 1)^2} \tag{5}$$

where $r$ is the linear correlation between observations and simulations, $\sigma$ is the standard deviation and $\mu$ is the mean.

Values $\text{NSE} > 0$ ($\text{NNSE} > 0.5$) are typically used as the benchmark to establish a model as a 'good' model ($\text{NSE} = 0$ indicates that the model is a predictor as good as the mean of the observations). The KGE equivalent values to consider that a model is skilful are $\text{KGE} > 1 - \sqrt{2} \approx -0.41$ (Knoben et al., 2019).

To estimate the effect of the new snow module on predictions of $\delta^{18}O_{\text{TRC}}$, we compared the parameters influencing $\delta^{18}O_{\text{TRC}}$ obtained by independent calibrations.

We also investigated the relative contributions to $\delta^{18}O_{\text{TRC}}$ of the source (xylem) water and of the fractionation processes during transpiration in the leaf (see the Eqs. (B1) and (B2) in Appendix B. Using the same approach as in Lavergne et al. (2017), we compared the predicted $\delta^{18}O_{\text{TRC}}$ from the reference simulations with those obtained from two experiments. First, to isolate the contribution of the source water on $\delta^{18}O_{\text{TRC}}$, we set the relative humidity ($h_{air}$) and $\delta^{18}O_V$ constant using the average values of $h_{air}$ and $\delta^{18}O_V$ obtained from the reference simulations. Second, to isolate the contribution of the isotopic enrichment of the leaf water during transpiration on $\delta^{18}O_{\text{TRC}}$, we set $\delta^{18}O$ in xylem water ($\delta^{18}O_{\text{XW}}$) constant using the average value of the reference simulation. We then compared the reference and experimental simulations using the coefficient of determination ($R^2$).

## 2.6 Validation of the snow model

The snow module was validated using a split-sample approach. At both of our study sites, we divided the period of SNDP observations (1979-2013) into half-periods: 1979-1996 and 1997-2013. We then used the SNDP observations in each half-period to calibrate the snow module, following the same procedure described in section 2.2 for the whole period of observations. Using the snow parameter obtained from each half-period, we simulated the whole period and compared the simulated snowpile with the snow observations using the NNSE and KGE metrics. To account for gaps in the observational record, we performed this comparison on the Mean Annual Cycle of the signal (calculated by averaging the series for every day of the year) which was also smoothed with a 30-day spline.

## 3 Results

### 3.1 Model validation

Using the split sample calibration method, MAIDENiso was able to simulate SNDP data for the full period that compared well with the SNDP observations at each site, as indicated by the two NNSE and KGE metrics (Table 3). The obtained values were well above the thresholds that establish the model as better predictor than the mean of the observations, with thresholds of NNSE=0.5 and KGE=0.41. In addition, the calibrations using half-periods produced similar results to the calibration using the full period, with the first half-period (1979-1996) producing slightly higher values than the second (1997-2013). The most notorious difference was the KGE obtained for the second half-period in Caniapiscau, which showed lower values (0.49) than the first half-period (0.69) and the full period (0.66), but it was still within the values needed to be considered as a good model result.

### 3.2 Snow calibration and impact on hydrology

MAIDENiso simulated SNDP and snow density using NARR meteorological data at our two sites, which we compared with the SNDP product from the NARR dataset. To compare with the direct observations of SWE at the Caniapiscau site, we used the simulated snow density to transform the SNDP (both that simulated by MAIDENiso and the NARR product) into SWE, which are shown in Fig. 3.

The MAIDENiso simulations reproduced the temporal change of the snowpile and showed a similar pattern to the real SWE observations collected at the Caniapiscau site (Fig. 3b). In contrast the NARR-based SWE estimates showed higher values of the snowpack during the winter months and offsets in the timing of snow accumulation and melting. The discrepancies between the NARR-based SWE data and the MAIDENiso SWE simulations could arise from a mismatch between the NARR meteorology (used to drive the model) and the NARR's snowpile data, which according to the available documentation the latter was artificially increased to match other sources (Mesinger et al., 2006). Therefore, our SWE simulations made by MAIDENiso using as inputs NAAR meteorological data were in better agreement with the direct observations of SWE at the Caniapiscau site than the SWE data obtained directly from the NARR dataset.

The calibration process converged and constrained the values of $r_a$ (resistance to water vapor transfer) well at both sites, as shown in Table 4. We obtained a value of $r_a$ almost twice as large at Tungsten ($87.35\,\mathrm{s\,m^{-1}}$) than at Caniapiscau ($47.88\,\mathrm{s\,m^{-1}}$).

Figure 4 shows the combined runoff and drainage simulated by MAIDENiso with and without including the snow module and the observations of river discharge in the Caniapiscau basin (scaled by the area of the basin). The observations showed a peak in river discharge between May and July, corresponding to the melting of the snow accumulated during winter. In the simulations computed using the original version of MAIDENiso (without snow module) the outflux of the model resembled the pattern of precipitation during the year because all incoming precipitation was considered as liquid and did not show any peak. Conversely, the simulations produced using our new MAIDENiso version (with snow module) did not have any outflux during winter, when all water is in solid state, and reproduced more accurately the peak of water outflux during spring melting. Overall, the timing of the spring discharge was well reproduced by the MAIDENiso version with snow, while the original

version was unable to simulate this peak. This improvement was confirmed by the NNSE between the observed and modelled river discharge at Caniapiscau which was lower for the model without snow (NNSE = 0.1) than for the new model with snow (NNSE = 0.45). The KGE for the observed and modelled river discharge at Caniapiscau improved only from -1.04 (without snow) to -0.57 (with snow), which indicates that our modelled river discharge (instant additions of drainage and runoff from all of the basin) can still be further improved.

### 3.3 GPP calibration

GPP was calibrated twice at each station: first for the original version of MAIDENiso and second for the new version with snow. Both versions were able to predict GPP observations in a similar way for both the Uaf and the EOBS stations (Fig. 5). The observed and simulated GPP were in good agreement regarding the timing (onset and offset) of the yearly peaks. The maximum of these peaks was higher for the observations, but this is due to exceptional days of very high observed GPP. Overall, the average GPP during the growing season was well reproduced at both GPP stations.

A Welch t-test determined that the GPP simulated by both versions of the model were statistically not different at the two sites of EOBS (t=0.04, p-value=0.97) and Uaf (t=0.47, p-value=0.63). This lack of influence of the snow module in the GPP was expected considering that while the snow module increases the availability of water in spring, the trees in our sites are not limited by water availability. This result also guarantees that the effects of the snow module on the $\delta^{18}$O ouputs that we are testing below are not affected by GPP by means of Eq. (B4).

### 3.4 Effects of the snow module on the $\delta^{18}$O outputs

Both model versions (MAIDENiso with and without snow) reproduced the mean level of the $\delta^{18}$O$_{\mathrm{TRC}}$ series (Fig. 6). This was expected because 1) the calibration process maximized the coincidence between observed and simulated $\delta^{18}$O$_{\mathrm{TRC}}$, and 2) the biochemical and kinetic fractionation parameters $\epsilon_0$ and $\epsilon_k$ could compensate the mean level of $\delta^{18}$O$_{\mathrm{TRC}}$ in Eq. (B1) for differences in $\delta^{18}$O$_{\mathrm{XW}}$. Regarding the agreement between inter-annual variations, the observed and simulated $\delta^{18}$O$_{\mathrm{TRC}}$ were not significantly correlated when snow was absent in the model (Fig. 6), but they were significantly correlated when MAIDENiso included snow (r = 0.57 in Tungsten and r = 0.52 in Caniapiscau, p < 0.01; Fig. 6). In the case of Caniapiscau, none of the versions of the model were able to simulate the amplitude of the variability of the observed $\delta^{18}$O$_{\mathrm{TRC}}$, which makes difficult to appreciate the improvement in the correlation between simulated and observed $\delta^{18}$O$_{\mathrm{TRC}}$ when adding snow to the model in Fig. 6b. To facilitate the interpretation of Figs. 6a,b, we have standardized the observed and simulated $\delta^{18}$O$_{\mathrm{TRC}}$ series (transformed to mean 0 and standard deviation 1) in Figs. 6c,d.

The distribution of the optimized parameters controlling $\delta^{18}$O$_{\mathrm{TRC}}$ can help to assess how the model compensated for the absence of snow (Table 4 and Fig. 7). In our simulations, adding snow induced an increase in the dampening factor $f_0$ at both sites, suggesting that the signal of the source water on $\delta^{18}$O$_{\mathrm{TRC}}$ was stronger than without considering snow.

To cast light on the effect of the snow module on $\delta^{18}$O, we compared the $\delta^{18}$O values from various parts of the model (precipitation, xylem water, discharge water, leaf water and TRC), for both versions (with and without snow) of MAIDENiso at the two study sites (Fig. 8). The $\delta^{18}$O of the precipitation ($\delta^{18}$O$_{\mathrm{P}}$, Fig. 8a) is the original source of the isotopic signals in the

other parts of the model and it matches well to the IsoGSM data we used for its calibration, although it is consistently lower than the monthly data of the OPIC (Online Isotopes in Precipitation Calculator) at both sites. The snow directly impacted the $\delta^{18}O$ of the source (xylem) water, $\delta^{18}O_{XW}$ (Fig. 8b). Without the snow module, $\delta^{18}O_{XW}$ followed closely the $\delta^{18}O_P$ signal shown in Fig. 8a, with a small delay due to the isotopic mixing in the soil. The $\delta^{18}O_{XW}$ signal was slightly enriched respect to $\delta^{18}O_P$ due to isotopic fractionation associated to soil and canopy evaporation. In contrast, when snow was present in the model, the soil absorbed melted water from the snowpile in spring, which was enriched due to fractionation during sublimation in winter. The $\delta^{18}O_{XW}$ values were higher in the version with snow. The difference in $\delta^{18}O_{XW}$ values between the two versions reduced in time due to the infiltration and mixing of the enriched summer precipitation. Isotopic composition of discharge water ($\delta^{18}O_{dis}$, Fig. 8c) follows a similar pattern to xylem water but with a delay of about two months, and lower amplitudes in the seasonal variations.

The daily $\delta^{18}O$ in the leaf ($\delta^{18}O_{leaf}$) and the TRC calculated with Eqs. (B1) and (B2) are shown in Figs 8d and 8e. The existence of the snow layer induced a higher $\delta^{18}O$ at the leaf level, especially in spring, due to the higher $\delta^{18}O_{XW}$ level. The mean level of $\delta^{18}O_{TRC}$ did not change after adding snow, despite the enrichment of $\delta^{18}O_{leaf}$ and $\delta^{18}O_{XW}$, because it was compensated by the lower values of the biochemical fractionation $\epsilon_0$. A Welch t-test was used to test the hypothesis that the curves obtained for two versions of the model were different, which was confirmed in all figures.

## 3.5 Relative influence of xylem water and leaf-level processes to the $\delta^{18}O_{TRC}$ signature

Finally, we investigated the relative contributions from the source water through the $\delta^{18}O_{XW}$ and the leaf transpiration enrichment throughthe $\delta^{18}O_{leaf}$ on the $\delta^{18}O_{TRC}$ timeseries (Fig. 9, peak values in Table 5). At both sites, the leaf water $\delta^{18}O$ isotopic enrichment had a stronger influence on $\delta^{18}O_{TRC}$ than the $\delta^{18}O$ variability of xylem source water, as shown by the higher variance explained by the experiment that simulated $\delta^{18}O_{TRC}$ considering only the effect of $\delta^{18}O$ leaf water enrichment indicated by a higher coefficient of determination $R^2$.

The addition of snow increased the $R^2$ for both types of experiments, but more importantly for the xylem source water experiment. As a consequence, the difference between the $R^2$ of xylem and leaf experiments became smaller, although leaf transpiration still explained higher $R^2$ at both sites. This was in agreement with the increase in $f_0$ seen in Table 4 for both sites, pointing to an important influence of snow on the source water and on $\delta^{18}O_{TRC}$.

## 4   Discussion

In this study, we implemented a new snow module in MAIDENiso to simulate snowpack dynamics and improve the model representation of the soil hydrology and the isotopic fractionation of oxygen in water and tree-ring cellulose. In the following paragraphs we discuss the impacts of the snow module addition on the different components of the model (i.e. the hydrological, photosynthetic and $\delta^{18}O$ modules), addressing the skills and limitations of our approach. We also discuss the implications of our new snow module implementation in MAIDENiso for future studies.

## 4.1 Improvements of the hydrological module

The calibration of the snow module at our sites yielded a value of $r_a$ almost twice as large in Tungsten than in Caniapiscau. Because of the lack of data for daily wind speed, we chose to implement $r_a$ as a constant parameter. The higher value of $r_a$ at Tungsten implies that snow sublimated at a slower rate, likely associated to the average wind speed during winter at this site was smaller than at Caniapiscau, in accordance to the interpretation of $r_a$ (Lawrence et al., 2019; Blanken and Black, 2004).

The skills of MAIDENiso to reproduce the hydrological cycle improved with the implementation of the snow module. Because of the accumulation and melting of snow, the new version of MAIDENiso is now able to simulate the observed peak of river discharge in early spring, while the previous version without the snow module could not simulate any peak (Fig. 4). The magnitude of this peak cannot be compared directly with observations because downscaling the river discharge by the size of the basin is not enough to make a direct comparison, as we also need to consider the following. 1) The water outflux simulated by MAIDENiso is the surface runoff (which is incorporated immediately to the streams) plus the subterranean drainage (which takes a longer time to reach the stream), which creates a time difference between for outflux sources within the same spatial point. 2) The outflux from different points of the basin takes different times to reach the main stream of the basin. 3) The outflux over the whole area of the Caniapiscau basin is not necessarily identical. A routing model (Oki et al., 1999; Southworth et al., 2007; Miller et al., 1994) could be used to calculate the delay and flow to the main stream from across the whole basin, for both types of water outflows. The incorporation of a routing model in MAIDENiso would allow to produce an estimate of streamflow for a basin, allowing for direct comparison with river discharge observations.

## 4.2 No effects of snow on photosynthesis

The calibration process yielded two different sets of parameters when using the two versions of the MAIDENiso model but resulted in similar predicted GPP values. Th parameters controlling GPP wee obtained showed very similar posterior distributions and values in the plausible block (Table A1, Figs A1, A2). These similarities indicate that, at our study sites, photosynthesis was not very sensitive to additional water from snow melt, suggesting that radial tree growth was not limited by water availability. This is in agreement with previous studies showing that in high latitudes soil humidity is not often a major constraint on tree growth and trees are usually mostly sensitive to temperature (Boisvenue and Running, 2006; D'Orangeville et al., 2018). However, different results could be found in sites where trees are more dependent on water derived from snowmelt (Du et al., 2014). Because GPP is not affected by the snow module, our study sites are ideal to investigate the effects of snow on $\delta^{18}O_{TRC}$ variations because it allows us to discard GPP as a possible cause for the differences observed between the two model versions.

## 4.3 Effects of snow on xylem water, leaf and tree-ring cellulose $\delta^{18}O$

Following the approach proposed by Lavergne et al. (2017), we produced yearly $\delta^{18}O_{TRC}$ timeseries by weighting the daily values with the GPP, assuming that C allocation to the stem is proportional to GPP. MAIDENiso has a module for the allocation of available C to the different parts of the tree, which provides an alternative and more realistic way of calculating yearly $\delta^{18}O_{TRC}$. However, the calibration of this module ideally requires observations of the same units as the product of the

allocation module, i.e. C mass per unit of stand basal area allocated to the stem. Although TRW data was available for both sites, their use was complicated because TRW observations represent just a portion of the total C allocation of the entire tree and do not offer an intra-annual C allocation resolution to constrain the simulations. Therefore, the use of GPP to weight the daily $\delta^{18}O_{TRC}$ was the best option for this particular study.

The model without snow produced depleted values of $\delta^{18}O_{XW}$, because it lacked the enrichment effect of evaporative fractionation in snow. This was also reflected in the $\delta^{18}O_{dis}$ signal, with depleted values for the model without snow. This contradicts studies that show that stream water does not usually show signs of evaporative fractionation (Evaristo et al., 2015). However, the discharge in MAIDENiso is simply the addition of the runoff and the drainage from the upper soil layers, and lacks the complexity of interactions with deeper groundwater that the real soil has. Therefore, while orientative, $\delta^{18}O_{dis}$ is only useful as a possible input to more complex groundwater model.

The addition of the snow module corrects for an important overcompensation effect that stemmed from the unrealistic representation of hydrology in the previous version. The model without snow produced depleted values of $\delta^{18}O_{XW}$, which the model had to compensate through the $\delta^{18}O$ parameters (higher values of $\epsilon_0$ and $\epsilon_k$, and lower contributions of xylem water through lower $f_0$). The calibration of the $\delta^{18}O$ processes for the two versions of MAIDENiso and the two sites yielded significant differences in the optimized parameters, both in the distribution of the optimal blocks and the values in the plausible blocks (Fig. 7, Table 4). The dampening factor $f_0$, which controls the direct contribution of the source water to the $\delta^{18}O_{TRC}$ signal, was significantly higher (especially in Tungsten) after adding snow. The calibration of the model without snow converged to a value of $f_0 \approx 0.32$, with the posterior distributions pushing toward the lower prior limit of 0.3 (Table 4, Fig. 7), which suggests that the calibration procedure would have converged towards a smaller value if it had been allowed. Adding snow increased the dampening factor to $f_0 = 0.48$ in Tungsten and $f_0 = 0.43$ in Caniapiscau, in agreement with the range of $f_0 = 0.4 - 0.5$ reported in previous studies (Roden and Ehleringer, 2000; Saurer et al., 1997; Sternberg et al., 1986; Yakir, 1992). Lavergne et al. (2017) obtained a dampening factor of $f_0 = 0.41$ in Quebec using the original model because the parameters from Eq. (2) could be calibrated to compensate for the absence of snow. These findings indicate that the addition of snow allows the model to increase the contribution of the source water to $\delta^{18}O_{TRC}$.

The calibrated value for the biochemical fractionation $\epsilon_0$ was different at the two sites, ranging with snow to without snow from 27.41 to 27.85 ‰ in Tungsten and from 24.15 to 24.48 ‰ in Caniapiscau (Table 4). The $\epsilon_0$ values were slightly higher at both sites when the model lacked snow, which suggests that the calibration compensates for consistently lower values of $\delta^{18}O_{XW}$ and/or $\delta^{18}O_{leaf}$ in Eq. (B1) to adjust the mean $\delta^{18}O_{TRC}$ to the observations. The kinetic fractionation $\epsilon_k$ obtained also differed strongly between sites, with snow to without snow from 11.8 to 13 ‰ at Tungsten and from 22.8 to 23.4 ‰ at Caniapiscau (Table 4). The $\epsilon_k$ was set to 26.5 ‰ by Farquhar et al. (1989), but it can vary over a larger range (Buhay et al., 1996). Lavergne et al. (2017) obtained a value of $\epsilon_k = 17.20$ ‰ for Quebec, with a similar posterior distribution that the one obtained here.

Our results also showed that the leaf $^{18}O$ enrichment due to transpiration has a stronger influence on $\delta^{18}O_{TRC}$ than the isotopic composition of the source (xylem) water, both in Tungsten and Caniapiscau, suggesting that it is the main driver of $\delta^{18}O_{TRC}$ variations. These results are in agreement with Lavergne et al. (2017) findings and reflect the strong effect of

vapor pressure deficit on $\delta^{18}O_{leaf}$ in Quebec. Nevertheless, the $\delta^{18}O_{TRC}$ signature also had a strong imprint of the source water signal as recently reported for the Tungsten $\delta^{18}O_{TRC}$ record that shared the same large-scale atmospheric patterns than spring-summer $\delta^{18}O_P$ (Field et al., 2021).

    The addition of the snow module to MAIDENiso therefore frees the calibration process from having to overcompensate for the artificially depleted $\delta^{18}O_{XW}$ values during the growing season (Fig. 8b). As our results have shown, this significantly

increased the correlation between the observed and simulated $\delta^{18}O_{TRC}$ compared to the version without snow (Fig 6; r = 0.52 for Caniapiscau and r = 0.57 for Tungsten, p < 0.01 versus non-significant, respectively). The improvement of the predictive skill of the model with the snow module reflects the influence of winter precipitation on physiological processes. Without snow, all winter precipitation passes through and out of the hydrological system without affecting the trees. In contrast, including snow allows winter precipitation to affect $\delta^{18}O_{TRC}$ indirectly through the source water.

Overall, the improvements found in the $\delta^{18}O_{TRC}$ simulations at both sites indicate that snow plays a critical role in $\delta^{18}O$ of the source water and thus on the final signature of $\delta^{18}O_{TRC}$. Even if the addition of snow would not had resulted in a significant improvement of the correlation between the simulated and observed $\delta^{18}O_{TRC}$, accounting for snow-related processes along the mechanistic chain is necessary for the application of a process-based model in an environment where snow is present. Process-based models are useful to understand complex processes, and while they may not necessarily produce better simulations

(closer to observations) than response functions, they can be calibrated under favorable conditions and then used for different datasets (Guiot et al., 2014). The incorporation of the snow module in MAIDENiso is therefore required for predicting tree-ring isotopic composition in forests located in cold environments where snow is present.

### 4.4    Implications for future studies

    Based on our results and comparison with other studies, we can conclude that the snow module predicted more realistic

and robust fluxes of water within the soil-plant-atmosphere continuum. The improvement of MAIDENiso to disentangle the contribution from the source (xylem) water and the $\delta^{18}O_{leaf}$ enrichment signal on $\delta^{18}O_{TRC}$ can help to track the origin of the isotopic signal and eventually improve the interpretations of the climate signal recorded in the tree rings. This is important because tree-ring isotopes are important climate proxies (Cernusak and English, 2015). The inclusion of the new snow module in the model can provide a more accurate representation of the physical and physiological processes taking place than in earlier

studies that did not take into account the additional effects of snowpack dynamics on $\delta^{18}O_{TRC}$, e.g. Lavergne et al. (2017). Now, MAIDENiso can simulate more reliable interactions between the coupled water and carbon cycles and tree physiological mechanisms in cold environments. Our findings will contribute to reduce uncertainties in the predictions of the response of forest productivity to hydrological changes, leading to better forward predictions that can eventually be used to reconstruct seasonal and long-term hydroclimatic variations.

An inverse modelling approach has previously been developed and tested using MAIDENiso to reconstruct paleoclimate from tree-ring data in the Fontainebleau Forest, France (Boucher et al., 2014). However, this exercise was restricted to the reconstruction of meteorological variables during summer, and to regions where the tree-ring proxies were not significantly affected by winter meteorology. The inclusion of snow in the model opens new possibilities for reconstructing hydroclimate in

cold regions, considering that the new version of MAIDENiso produces simulation of $\delta^{18}O_{TRC}$ that account for snow-related
processes.

Suitable regions for the application of MAIDENiso in future studies include high-mountain regions, now that the model has
a working snow module. Regions with snow-dominated winters and dry summers, such as the South Western USA or some
Mediterranean sites, can also be of interest for future studies with MAIDENiso, as trees in these sites can be more dependent
on water derived from snow melt than the sites used in the present study. The application of MAIDENiso to any site is possible
provided that there is sufficient meteorological data to drive the model and local information to calibrate the model parameters
(GPP, snow, TRW and TRC stable isotopes). We expect MAIDENiso to be applied more broadly in high latitude and altitude
environments in the near future.

## 5    Conclusions

In this paper we presented the new snow module incorporated into MAIDENiso, which consists of new hydrological calcu-
lations of snow dynamics and a thermal module. Our results show how this snow module improves the simulation of outputs
associated with the hydrological cycle at cold and high latitude sites without affecting simulations from the carbon cycle
component. These findings were expected because GPP and tree-ring growth at the studied boreal high-latitude sites are not
constrained by soil moisture availability but by surface air temperature and light (Jarvis and Linder, 2000). The simulations of
the new version of MAIDENiso reproduce the observed $\delta^{18}O_{TRC}$ better than the original snow-less version of MAIDENiso.
Based on the development presented here, the potential for the application of MAIDENiso is notably increased.

*Code and data availability.*  The MAIDENiso code is available in the Zenodo repository. Note that there are two versions of the code, cor-
responding to the model with snow (https://doi.org/10.5281/zenodo.5597877) and the model without snow (https://doi.org/10.5281/zenodo.
5598076). In addition, a university website has been created (https://dendro-eco.uqat.ca/maiden/) for the MAIDEN model, where a technical
description of the model and access to different model versions will be available. The meteorological input files and parameter files needed
to run MAIDENiso and the observational data are available in the Zenodo repository (https://doi.org/10.5281/zenodo.5599091).

## Appendix A:  Photosynthesis model

GPP $(\mathrm{g\,C\,m^{-2}\,day^{-1}})$ in MAIDENiso derives from a coupled photosynthesis-stomatal conductance system. The leaf photo-
synthesis is modelled following Farquhar et al. (1980), scaled to the canopy followingDe Pury and Farquhar (1997) as explained
in Misson (2004). Daily Vcmax (Vcmax$_i$) is modelled as:

$$\mathrm{Vcmax_i} = \frac{\mathrm{Vmax}}{1 + \exp(\mathrm{Vb} \cdot (\mathrm{Tday_i} - \mathrm{Vip}))} \tag{A1}$$

The parameter Vmax determines how daytime temperature Tday controls the maximum carboxylation rate at day i. Because
there was no explicitly known mechanistic formula relating Vcmax and Tday, three parameters were introduced to control this

relationship in a non-linear way, i.e. Vmax, Vb and Vip. These parameters control the asymptote, the slope and the inflection point of Vcmax, respectively, and have to be calibrated.

The stomatal conductance for carbon ($\mu mol\ m^{-2}\ s^{-1}$) is calculated using the Leuning et al. (1995) model, modified by Gea-Izquierdo et al. (2015) to incorporate soil water stress:

$$g_{sc} = g_0 + g_1 \frac{A_n}{(C_a - \Gamma^*)(1 + VPD/VPD_0)} \theta_g\ P_{atm} \tag{A2}$$

where $g_0 = 0\,\mu\text{mol m}^{-2}\,\text{s}^{-1}$ and $g_1 = 10\,\mu\text{mol m}^{-2}\,\text{s}^{-1}$ are fitted parameters representing the residual conductance as the net assimilation rate ($A_n$) approaches zero and the slope of the function, respectively. $P_{atm}$ is the atmospheric pressure (Pa). $C_a$ is the atmospheric $CO_2$ pressure (Pa). $\Gamma^*$ is the $CO_2$ compensation point in the absence of dark respiration (Pa), which is calculated following Bernacchi et al. (2001). VPD is the vapor pressure deficit (kPa), and $VPD_0$ is an empirically fitted parameter representing the sensitivity of stomata to changes in VPD (usually around 15 kPa; Knauer et al. (2015)). $\theta_g$ is the empirical soil water stress factor, a non-linear function ranging between 0 when the soil is too dry for the roots and 1 in absence of water stress:

$$\theta g = \frac{1}{1 + \exp(soilb \cdot (SWC_i - soilip))} \tag{A3}$$

The water stress level depends on the soil water content (SWC, mm), but the current version of MAIDENiso lacks a mechanistic model to explain the relationship between soil water content and water stress. For this reason, this relation is modelled as a logistic function, introducing the calibration parameters soilb and soilip as the slope and the inflexion point of $\theta g$.

Finally, there is a time lag between the recovery of photosynthesis and the temperature increase in spring that is taken into account by the model. This is done by replacing Tday in Eq. (A1) by the temperature transformation S, defined as:

$$\frac{dS_i}{di} = \frac{Tday_i - S_i}{\tau} \tag{A4}$$

where $\tau$ is a parameter representing the number of days needed by the tree to adapt the photosynthesis to changing temperatures.

There are a total of 6 undetermined parameters that control GPP production in MAIDENiso in Eqs. (A1), (A3) and (A4). These parameters were calibrated at the two eddy covariance flux stations described in section 2.4, for both versions of the model. These parameters, their prior distributions and their posterior distributions are shown in Table A1. For better visualization, the probability distribution function (pdf) of the posterior distributions of the GPP parameters are also shown in Figs. A1 and A2.

**Appendix B: Isotopic model**

MAIDENiso keeps track of the stable isotopic composition of oxygen ($\delta^{18}O$) in all the water/ice pools and fluxes of the hydrological model (Fig. 2). The isotopic module calculates fractionation from evaporation (from soil and canopy water) and transpiration at leaf level to produce an isotopic oxygen signature in TRC ($\delta^{18}O_{TRC}$). This is based on the Danis et al. (2012)

formulation of the Craig-Gordon model (Craig and Gordon, 1965):

$$\delta^{18}O_{TRC} = (1 - f_0) \cdot \delta^{18}O_{leaf} + f_0 \cdot \delta^{18}O_{XW} + \epsilon_0 \tag{B1}$$

With $\delta^{18}O$ at leaf level being:

$$\delta^{18}O_{leaf} = \epsilon^* + \epsilon_k \cdot (1 - h_{air}) + h_{air} \cdot \delta^{18}O_V + (1 - h_{air}) \cdot \delta^{18}O_{XW} \tag{B2}$$

Here, $f_0$ (unitless) is the dampening factor reflecting the exchange of the oxygen atoms between sucrose and xylem water during the synthesis of cellulose in the xylem cells of the tree rings, typically within a range of 0.4-0.5 (Roden and Ehleringer, 2000; Saurer et al., 1997; Sternberg et al., 1986; Yakir, 1992). $\epsilon_0$ is the biochemical fractionation due to oxygen exchange

between water and the carbonyl groups (C=O) in the organic molecules, undetermined but expected in a range of 24-30 ‰ (DeNiro and Epstein, 1979; Farquhar et al., 1998). $\epsilon^*$ is the equilibrium fractionation due to the change of phase of water from liquid to vapor at leaf temperature (fixed at 21.4 °C, the temperature threshold for maximum carbon assimilation), with a value of 9.65 ‰ (Helliker and Richter, 2008). $\epsilon_k$ is the kinetic fractionation due to the diffusion of vapor into unsaturated air through the stomata and the leaf boundary layer, set to 26.5 ‰ in Farquhar et al. (1989) but we consider it undetermined because it can

vary over larger ranges (Buhay et al., 1996). $h_{air}$ is the relative humidity, which is estimated in MAIDENiso from the daily air temperature and the dew point temperature (Running et al., 1987). $\delta^{18}O_V$ and $\delta^{18}O_{XW}$ are the $\delta^{18}O$ of vapor and xylem (source) water, respectively. $\delta^{18}O_V$ is calculated from the $\delta^{18}O$ of precipitation ($\delta^{18}O_P$) and the fractionation due to the phase change from liquid water to vapor at mean air temperature, $\epsilon^*_{T_{air}}$ (Horita and Wesolowski, 1994):

$$\delta^{18}O_V = \delta^{18}O_P - \epsilon^*_{T_{air}} \tag{B3}$$

The $\delta^{18}O_{TRC}$ time series produced through Eq. (B1) are daily, while the $\delta^{18}O_{TRC}$ measured from tree rings is commonly annually resolved, or occasionally with intra-annual resolution (e.g. Szejner et al. (2018)). To produce a yearly record comparable with observations, the daily series are weighted with GPP. This assumes that allocation of carbon to the trunk is proportional to daily GPP ($GPP_d$):

$$\delta^{18}O_{TRC,y} = \frac{\sum_d \delta^{18}O_{TRC,d} \cdot GPP_d}{\sum_d GPP_d} \tag{B4}$$

*Author contributions.*  I.H.M. implemented the new snow module into MAIDENiso, performed the simulations and analyses and wrote the first draft of the manuscript. F.G. and A.L. helped in setting up the new module into the original version of the model. L.A.-H, R.F. and E.B. provided the $\delta^{18}O_{TRC}$ chronology. E.B., L.A.-H, F.G. and A.L. contributed to the design of the study (analyses to perform and structure the paper), and interpretation of the results. All authors contributed to improve the manuscript and guided the simulations and analyses.

*Competing interests.*  The authors declare that they have no conflict of interest.

*Acknowledgements.* This research was supported by the Natural Sciences and Engineering Research Council of Canada (NSERC) PERSIS-TENCE project (RDC 485475 - 15 to E.B.), and the US National Science Foundation (NSF) grants PLR-1504134, PLR-1603473, AGS-1502150 and OISE-1743738 awarded to L.A-H. The PERSISTENCE project is a collaborative research grant that involves the participation of Hydro-Québec, Manitoba Hydro and the Ouranos consortium. We thank L. Perreault and D. Tapsoba from Hydro-Québec for providing the SWE data at the Caniapiscau basin used here. A.L was supported by a Marie Sklodowska-Curie Individual Fellowship under the European

Union's Horizon 2020 Research and Innovation Programme (grant agreement no: 838739 ECAW-ISO). F.G. was supported by the Ministère des Forêts, de la Faune et des Parcs (MFFP; contract number 142332177-D), and NSERC (Alliance Grant number ALLRP 557148-20). We are thankful to Wei Huang and Jean-Francois Hélie, respectively from the Stable Isotope Laboratory of Lamont-Doherty Earth and GEOTOP (UQAM), for their support with isotopic measurements.

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

| Parameter | Physical meaning | Parameter type | Units |
|---|---|---|---|
| $r_a$ | Air resistance to water vapor transfer | Free | $\text{s m}^{-1}$ |
| $a_{snow}$ | Slope of the linear temperature dependence | Site | $\text{‰ °C}^{-1}$ |
| $b_{snow}$ | Slope of the linear precipitation dependence | Site | $\text{‰ mm}^{-1}$ |
| $c_{snow}$ | Intercept of the linear model | Site | ‰ |

**Table 1.** New parameters introduced to MAIDENiso in the new version.

| Site | Precipitation type | a (‰ $^{\circ}$C$^{-1}$) | b (‰ mm$^{-1}$) | c (‰) |
|---|---|---|---|---|
| Tungsten | Snowfall | 0.4124 | -0.0631 | -16.4182 |
| | Rainfall | 0.4583 | -0.9909 | -16.26 |
| Caniapiscau | Snowfall | 0.4007 | -1.622 | -13.1279 |
| | Rainfall | 0.2654 | -0.3613 | -11.4665 |

**Table 2.** Parameters obtained for the regression models for snow and rainfall using the IsoGSM dataset at the Tungsten and Caniapiscau sites. The thresholds for the null hypothesis are NNSE=0.5 and KGE=0.41.

| Site | Statistic | 1979-2013 | 1979-1996 | 1997-2013 |
|------|-----------|-----------|-----------|-----------|
| Tungsten | NNSE | 0.69 | 0.67 | 0.64 |
| | KGE | 0.67 | 0.68 | 0.66 |
| Caniapiscau | NNSE | 0.73 | 0.73 | 0.72 |
| | KGE | 0.66 | 0.69 | 0.49 |

**Table 3.** Normalized Nash-Sutcliffe Efficiency (NNSE) and Kling-Gupta Efficiency (KGE) for the mean annual cycles of SNDP simulated by calibrating the model with the full period and the two half-periods, at the Tungsten and Caniapiscau sites.

| Parameter | Units | Prior range | Posterior - Tungsten Without snow | Posterior - Tungsten With snow | Posterior - Caniapiscau Without snow | Posterior - Caniapiscau With snow |
|---|---|---|---|---|---|---|
| $r_a$ | s m$^{-1}$ | 0/400 | | 85.91/90.50 (87.35) | | 44.94/50.83 (47.88) |
| $f_0$ | n/a | 0.3/0.5 | 0.30/0.42 (0.32) | 0.37/0.5 (0.48) | 0.30/0.43 (0.32) | 0.33/0.49 (0.43) |
| $\epsilon_0$ | ‰ | 24/30 | 25.43/28.42 (27.85) | 25.36/27.67 (27.41) | 24.03/26.36 (24.48) | 24.02/25.51 (24.15) |
| $\epsilon_k$ | ‰ | 10/30 | 10.91/25.15 (13.03) | 10.37/22.83 (11.81) | 14.07/26.52 (22.77) | 16.23/26.48 (23.41) |

**Table 4.** Calibration parameters for the snowpile (for the version with snow exclusively) and $\delta^{18}O_{\mathrm{TRC}}$. Parameters, units, prior range, and posterior range (with parameter value in the plausible block) for both MAIDENiso versions and both sites.

| Site | Model version | Xylem | Leaf |
|---|---|---|---|
| Tungsten | Without snow | 0.676 | 0.835 |
| | With snow | 0.832 | 0.940 |
| Caniapiscau | Without snow | 0.706 | 0.674 |
| | With snow | 0.711 | 0.757 |

**Table 5.** Mode of the probability density function for the coeficients of determination $R^2$ in Fig. 9 between the reference simulations and the water source (xylem) experiments and the leaf water enrichment experiments, for the model without snow and with snow and for the two sites of Tungsten and Caniapiscau.

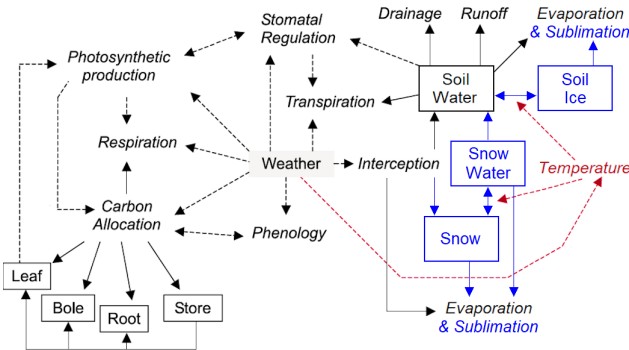

**Figure 1.** Diagram showing the main features in the new version of MAIDENiso, with old components and fluxes in black and new ones in blue for snow/ice and in red for the thermal module. Processes are in italics, boxes are carbon and water pools, broken lines are links between processes, and solid lines are carbon and water fluxes. Figure modified from Misson (2004).

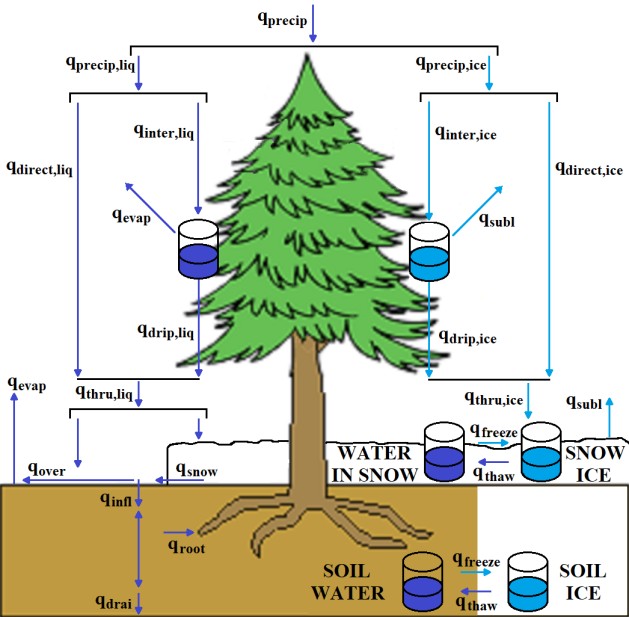

**Figure 2.** The hydrological system in the new version of MAIDENiso. Pools (flasks) and fluxes (arrows) are shown for liquid water in dark blue, and for snow/ice in light blue.

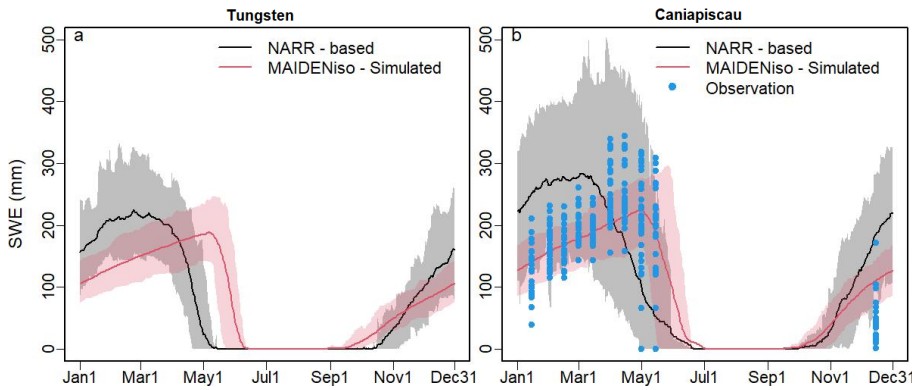

**Figure 3.** Snow Water Equivalent (SWE) averaged over 1979-1997 at the (a) Tungsten and (b) Caniapiscau sites, extracted from the NARR data (black) and simulated by MAIDENiso (red) using NARR meteorology. The solid line indicates the average of the same DOY during this period, and the shadows indicate the $2\sigma$ variability. Direct observations of SWE at Caniapiscau, taken at discrete intervals, are shown as blue dots.

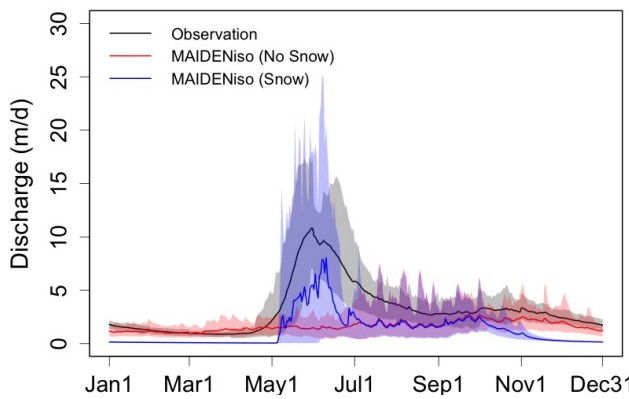

**Figure 4.** Water outflux (drainage + runoff) simulated for 1979-1997 in the Caniapiscau site by MAIDENiso (smoothed over a 10-day period), without (red) and with (blue) the snow module. The black line shows the discharge from observations from the Caniapiscau basin between 1979-1997, scaled with the area of the basin. Solid lines indicate the average of the same DOY during this period, shadows indicate the $2\sigma$ variability.

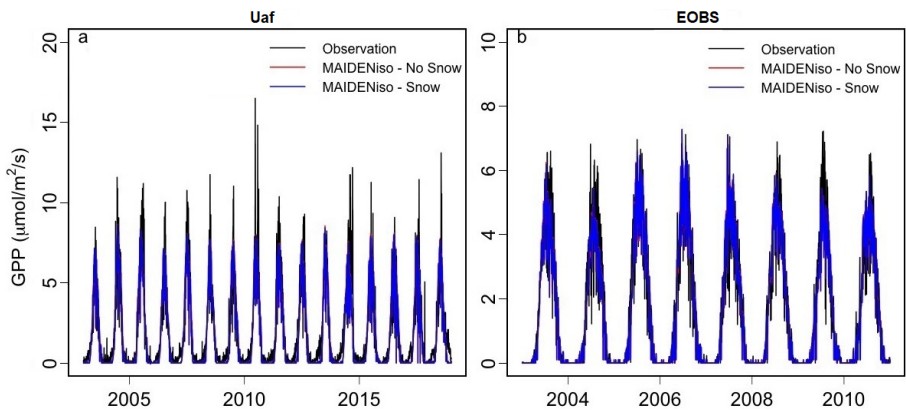

**Figure 5.** GPP at the Uaf site (a) and the EOBS site (b). We show observations from flux towers (black) and MAIDENiso simulations without (red) and with (blue) snow processes. Independent optimizations are run for the two versions of MAIDENiso. A t-test determined that the GPP simulated by the model with and without snow are statistically not different, for this reason the red line is overlapped by the blue line.

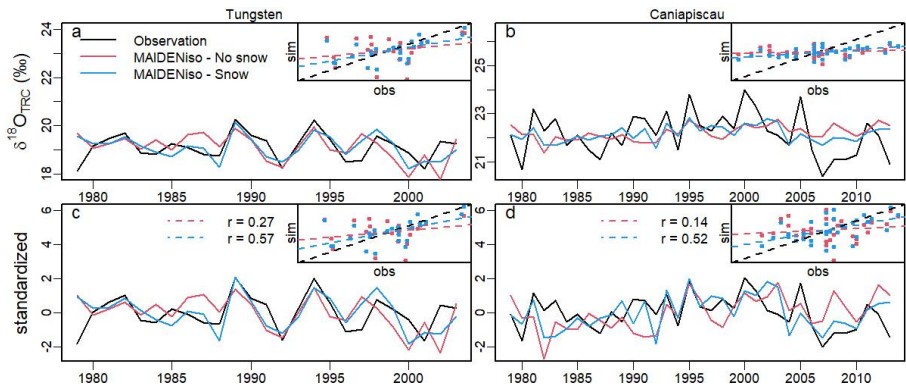

**Figure 6.** The $\delta^{18}O_{TRC}$ observed (black) and simulated by the versions of MAIDENiso without (red) and with (blue) snow, using the calibrated parameters for (a,c) Tungsten and (b,d) Caniapiscau. Top panels show the raw $\delta^{18}O_{TRC}$ series, bottom panels show the standardized $\delta^{18}O_{TRC}$ series. Top-right inner panels show the scatter diagrams of the simulated (by both versions of the model, red for the version without snow and blue for the version with snow) and observed values of $\delta^{18}O_{TRC}$, with dashed lines showing the linear regression models. Correlations are identical for the raw series (top panels) and the standardized series (bottom panels).

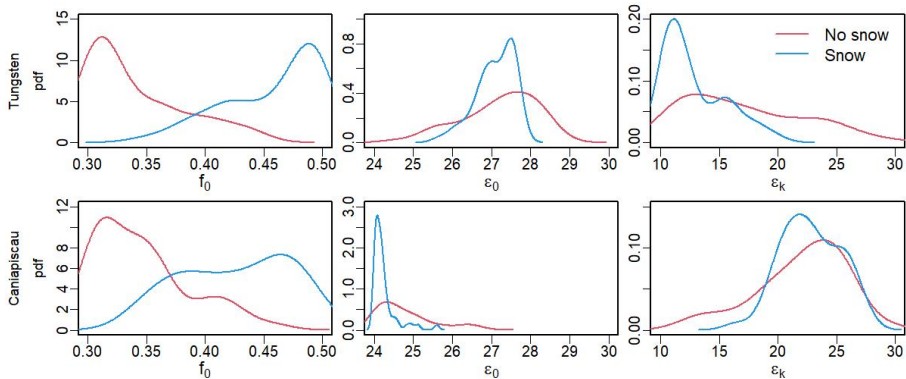

**Figure 7.** Posterior probability density distributions of the parameters controlling $\delta^{18}O_{\mathrm{TRC}}$ at the Tungsten and Caniapiscau sites, for the model without (red) and with (blue) snow.

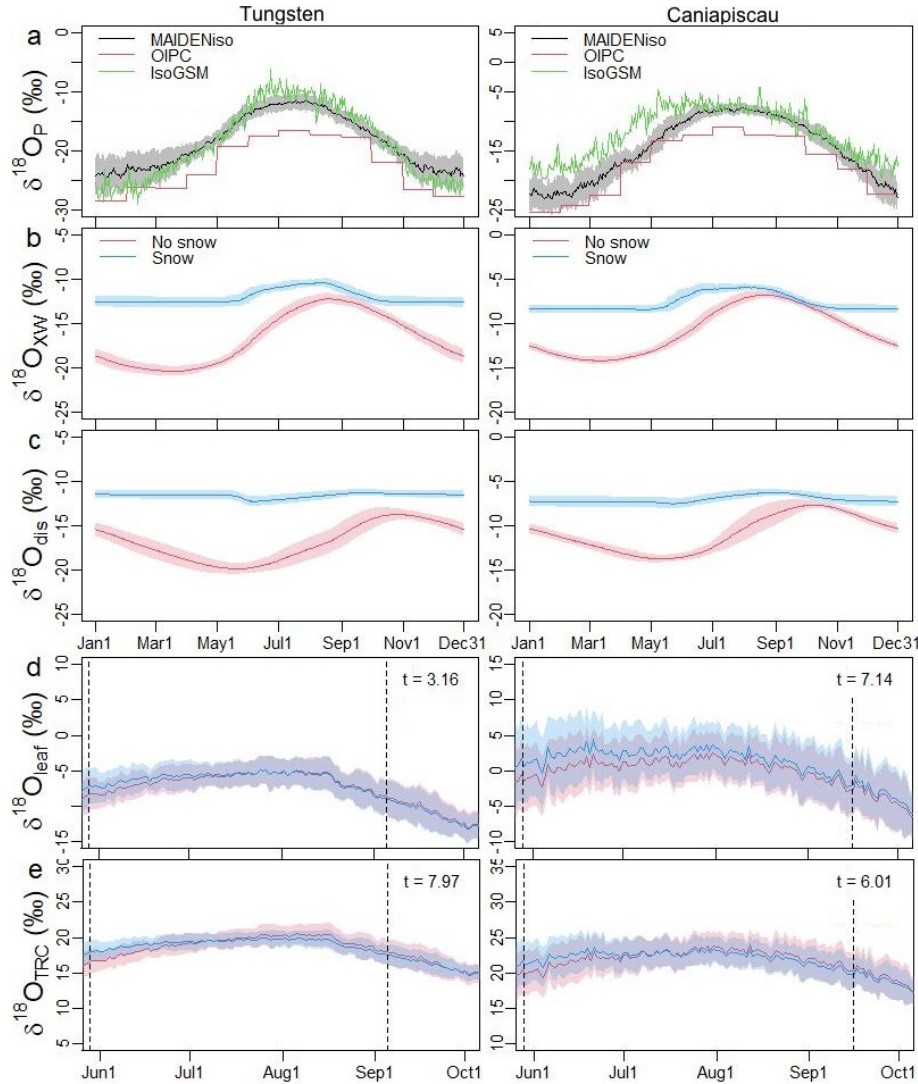

**Figure 8.** Simulated $\delta^{18}O$ at different stages of the water cycle for the period 1979-2003, for the sites of Tungsten (left) and Caniapiscau (right). a) The $\delta^{18}O$ in precipitation ($\delta^{18}O_P$) simulated by MAIDENiso using the NARR meteorology (black), monthly data from the Online Isotopes in Precipitation Calculator (OIPC, red) and mean from the IsoGSM dataset (green). b) The $\delta^{18}O$ in the xylem water ($\delta^{18}O_{XW}$), c) the $\delta^{18}O$ in the discharge water ($\delta^{18}O_{dis}$), d) the $\delta^{18}O$ in the leaf ($\delta^{18}O_{leaf}$) and e) in the Tree-Ring Cellulose ($\delta^{18}O_{TRC}$), for the model without snow (red) and with snow (blue). Shadows indicate the $2\sigma$ variability for the same DOY within the 1979-2003 period. Vertical dashed lines in d) and e) indicate the start (budburst) and end of the growth season. Note that isotopic calculations are still made outside of this period but no water is absorbed by the tree. d) and e) include the t-scores from a Welch t-test, which show that the curves obtained for two versions of the model are different.

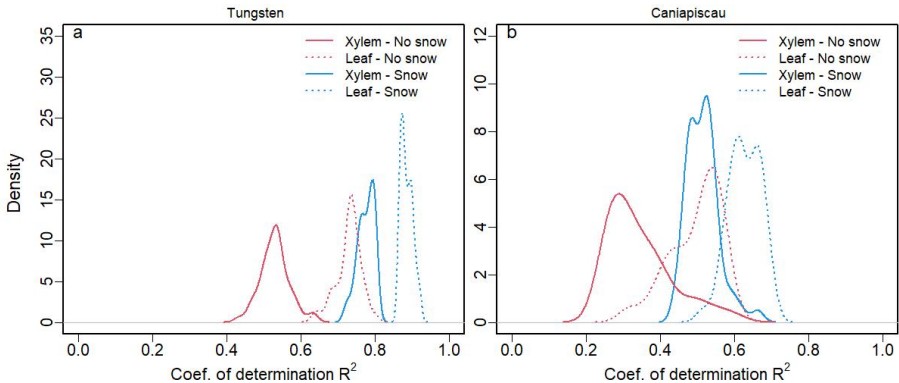

**Figure 9.** Density distribution of the coefficients of determination ($R^2$) between the reference simulations and the water source (xylem) experiments (solid line, $\delta^{18}O_V$ and $h_{air}$ set constant), and the leaf water enrichment experiments (dashed line, $\delta^{18}O_{XW}$ set constant), for the model without snow (red) and with snow (blue). Shown for a) Tungsten and b) Caniapiscau.

| Parameter | Units | Prior range | Posterior - EOBS Without snow | Posterior - EOBS With snow | Posterior - Uaf Without snow | Posterior - Uaf With snow |
|---|---|---|---|---|---|---|
| Vmax | $\mu$molC m$^{-2}$ s$^{-1}$ | 5/150 | 47/128 (59) | 47/125 (66) | 82/149 (141) | 82/147 (101) |
| Vb | n/a | -0.30/-0.10 | -0.22/-0.15 (-0.18) | -0.22/-0.15 (-0.17) | -0.25/-0.20 (-0.21) | -0.26/-0.20 (-0.22) |
| Vip | °C | 10/30 | 15.5/26.3 (18.2) | 15.8/26.1 (19.4) | 18.9/23.6 (22.8) | 18.7/23.4 (20.5) |
| soilb | n/a | 0.025/-0.005 | -0.023/-0.005 (-0.02) | -0.023/-0.006 (-0.013) | -0.023/-0.006 (-0.021) | -0.021/-0.006 (-0.021) |
| soilip | mm | 100/400 | 111/312 (179) | 120/260 (177) | 109/251 (161) | 102/273 (179) |
| $\tau$ | days | 1/20 | 12.7/17.1 (15.1) | 12.6/16.7 (15.1) | 13.8/17.1 (16.5) | 13.5/17.5 (15.0) |

**Table A1.** Calibration parameters for the GPP module. Parameters, units, prior range, and posterior range (with parameter value in the plausible block) for both MAIDENiso versions and both flux towers.

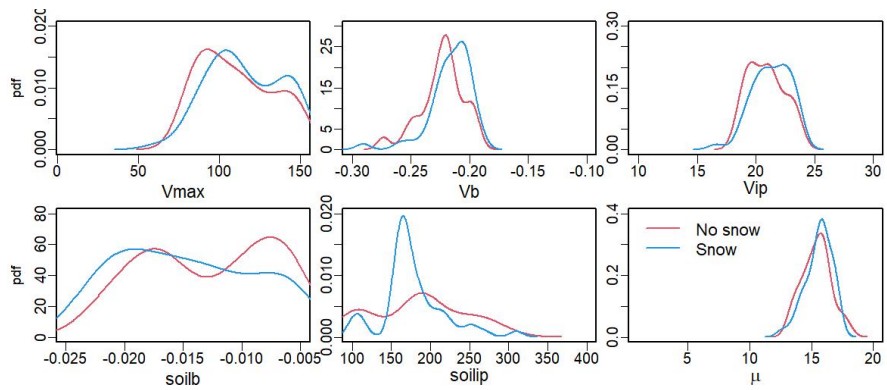

**Figure A1.** Posterior probability density distributions of the parameters controlling GPP at the Uaf site, for the model without (red) and with (blue) snow.

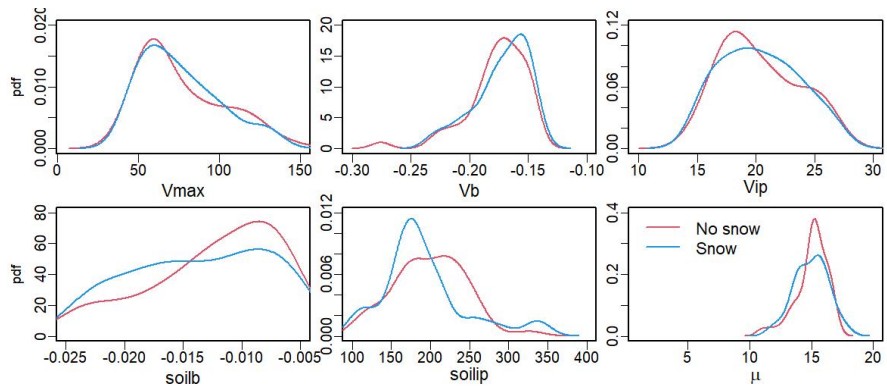

**Figure A2.** Posterior probability density distributions of the parameters controlling GPP at the EOBS site, for the model without (red) and with (blue) snow.