# Peer review of "A new snow module improves predictions of isotope-enabled MAIDENiso forest growth model"

_Geoscientific Model Development, 2021_

## Author Response (AR1)

Dear Editor,

We are very thankful to the reviewers for their comments, which we found constructive and appropriate. With the exception of the suggestion of expanding this study to the Mediterranean region, which may require additional improvements of the model that are out of the scope of this study, we have incorporated almost all of the reviewers' corrections and suggestions. We have included all these changes in the final version of the manuscript, and are detailed below point-by-point as answers to the reviewers' comments. In addition, we have corrected typographical mistakes and made many minor corrections throughout the manuscript to improve the readability of the text. Because these corrections are too numerous, we do not include them in a list to avoid making this cover letter tedious, but they can be easily seen throughout the tracked-changes version of the manuscript.

**General Comments and relevant changes:**

Both reviewers suggested performing an independent validation of the new snow module described in the manuscript to determine its predictive skills. We therefore calibrated the model over half of the period (either 1979-1996 or 1997-2013) and validated the calibration based on a comparison of simulations and observations for the whole period. We used the Nash-Sutcliffe model Efficiency (NSE) and the Kling-Gupta Efficiency (KGE) criteria to determine the agreement between simulated and observed snow depth (SNDP). While we considered performing this procedure as well for the $\delta^{18}O_{TRC}$ series, we discarded the idea on the basis of 2 arguments: 1) The $\delta^{18}O_{TRC}$ series is yearly and calibrating with only half of the period gives us very few points of data to use in the calibration. 2) The $\delta^{18}O_{TRC}$ module has not been modified since the previous version (which uses the implementation of Danis et al. (2012)), therefore it does not require further validation.

These calibration-validation experiments show that the model is able to forecast snow correctly outside of the calibration period, and that the simulated snow is consistent for different calibration periods. We believe this independent validation of the model addresses the main concerns of the reviewers, and it has been included in the revised version of the manuscript, along with other minor corrections addressing the specific comments of the reviewers.

Taking into account all the comments from the reviewers, we have made a number of significant changes to the manuscript, which we list here:
- As per request of the author LAH, she has been listed as 6th author instead of 5th author.
- A new subsection 2.6 "Validation of the snow model" has been added to the Materials and methods section.
- A new subsection 3.1 "Model validation" has been added to the Results section.
- A new appendix B "Isotopic model" has been added.
- Three new tables (Tables 1, 2 and 4) have been added.
- Two new subfigures (row c) have been added to Figure 8.

We now address the specific comments of the reviewers and public comments, pointing out the specific location of our changes in the revised manuscript.

**Specific comments:**
**Community comment from Ru Huang:**
*Line 30*, *in the introduction, the authors had summarized snow-related studies from the North America. I suggest to add snow-related studies in the Karakoram and Himalayas (e.g., Huang et al., 2019; Zhu et al., 2021, https://doi.org/10.1016/j.epsl.2018.11.002, https://doi.org/10.1007/s00382-021-05736-6)*

We thank the commenter for these suggestions. We have added them to the Introduction (page 2, line 31) for a more complete list of examples of snow-related studies.

*Line 220*, *I am wondering the climatic signals of tree-ring oxygen isotopes. As shown by Field et al. (2021), tree-ring oxygen isotopes from Tungsten is a proxy of springsummer temperature. How about the tree-ring oxygen isotopes from Caniapiscau? Do tree-ring oxygen isotopes from Caniapiscau also indicate temperature? Would it possible for the authors to add some discussions about the climatic signals in tree-ring oxygen isotopes in the manuscript?*

In response to this comment, we have looked at the correlations between tree-ring stable oxygen isotopes and the meteorology (minimum and maximum temperature and precipitation). Figure 1 shows the correlation of the $\delta^{18}O_{TRC}$ series with the meteorological variables (precipitation and maximum and minimum temperature) from NARR averaged over a moving 31-day window.

Field et al. (2021) found that the $\delta^{18}O_{TRC}$ signal at Tungsten is a proxy of spring (MAM) and summer (JJA) maximum daily temperature from the 1932-2002 data of the GHCN (Global Historical Climatology Network). We have used 1979-2003 data at Tungsten and 1979-2013 at Caniapiscau from the NARR (North American Reanalysis) dataset. Looking at Tungsten, our dataset is different and the period shorter than in Field et al. (2021), for this reason we have found the $\delta^{18}O_{TRC}$ signal at Tungsten to be a proxy of summer (JJA) maximum daily temperature. Both sites respond similarly to temperatures, with correlations peaking in the late season: summer in Caniapiscau and late spring and early autumn in Tungsten. Correlations with precipitation are weaker and differ slightly between sites: in Caniapiscau $\delta^{18}O_{TRC}$ is correlated negatively with mid-summer precipitation, while in Tungsten correlations are weak during this period.

This paper however does not aim to discuss the climate signals at these sites, but to introduce the new snow module to MAIDENiso and the consequences it has on the other simulated products (water discharge, GPP and stable oxygen isotopes). On the other hand, adding this analysis to the study does not add value to the study while it may distract the reader from the purpose of our study. Therefore, we decided to not include it in the manuscript. Instead, we have added basic information about the average meteorological variables (summer temperature, yearly precipitation and fraction of solid precipitation) at both sites from the datasets used in our study (page 7, line 194-198), which we found to be an information of interest to the reader..

[Figure]

Figure 1. Correlation between the mean of the series of observed δ18OTRC and monthly climate variables of the study areas (precipitation in black and maximum and minimum temperature in red and blue, respectively). For the climate variables, time windows of 31 days are used to obtain time series of monthly data (over the periods of 1979-2003 for Tungsten and 1979-2013 for Caniapiscau) for each day (central day), averaging the values of each window and each year.

*Lines 392-394, could the authors quantify the contributions of leaf enrichment due to transpiration and source water on tree-ring oxygen isotopes?*
These contributions were studied in subsection 3.5 "Relative influence of xylem water and leaf-level processes to the d18OTRC signature". Figure 9 represents the relative contributions of xylem water and leaf transpiration (given by position of the peaks in the X-axis, the coefficient of determination). However, we had not specified the numerical values that quantify these contributions. In order to provide more information to the reader, we have added a new table (Table 5) with these values.

*Line 496, the species Latin names for the species should be italic. Please check other part of the manuscript.*
We are thankful for this correction. All Latin names are now in italic in the text. We have also made sure that the full latin names for black spruce and white spruce are written.

*In Line 411, "4.4 Implications for future studies". It is better to extend the current study to other high-mountains regions (e.g., Alps, Himalayas, Andes Mountains) in the future.*
We agree that the high-mountain regions are suitable for the application of MAIDENiso, however the calibration of the model is made possible in our sites due to the availability of critical data for the snow-TR modelling: high-resolution weather records, snow measurements, isotopic tracking, dendrochronological measurements, flux measurements, etc. To apply it in different, highly-variable settings, we first need to identify sites that gather as many of these parameters as possible. The application of MAIDENiso to high-mountains is something we have put into consideration for future studies. We have indicated this in a new paragraph in the "Implications for future studies" subsection in the Discussion (pages 15-16, lines 481-487).

**Specific comments from Reviewer #1:**

***Section 50***: The authors wrote: *"Among the previously mentioned models, the only exception is the Biome3 model (Rathgeber et al., 2003)..."* It is not correct! As an example, the VS-model has a sub-block of snow melting (see Vaganov et al., 2006; Shishov et al., 2016). Could the authors introduce additional references there?

The reviewer is absolutely right: During our bibliographical study we missed the most recent versions of the VS model, which in fact includes snow. The physical modelling of snow in the VS model is, however, a basic model which assumes that accumulation and melting of snow is driven by air temperature, similar to the Biome3 model. We have included the references suggested by the reviewer (page 2, line 47) and we have acknowledged the snow model present in the VS model in the Introduction (page 2, line 52).

***Paragraph 2.1.1***: I would suggest to make the Paragraph 2.1.1 shorter, and simplify it by moving some formulas and description into appendix section. The well-known MAIDENiso is not a topic of the MS discussion.

As suggested by the reviewer, we have moved the description of the isotopic module to a new appendix section (appendix B: Isotopic model, pages 17-18). We have kept the descriptions of the model's hydrology in the main text because the purpose of the section was to highlight the deficiencies of the previous hydrological module that we are addressing in the new version of the model presented here. Overall, the length of text in subsection 2.1.1 has decreased from 50 to 25 lines (pages 3-4).

***Paragraph 2.1.2:*** How many new parameters involved in the soil moisture block? I would suggest to introduce a table there with short description of new parameters (included their dimensions).

We have added a new table (Table 1) describing the 4 new parameters added in the new version. This includes their description, their dimensions, and the type of parameter. The type of parameter can be a site parameter, which is determined externally to MAIDENiso, or a free parameter, which has to be determined by calibrating MAIDENiso. One of the 4 new parameters is a free parameter (i.e. resistance to water vapor transport), while the other 3 are site parameters. We have explained this in a new paragraph at the end of subsection 2.1.2 (page 6, lines 152-160).

***Section 170:*** Is the new code of the MAIDENnIso available in some public depositories?

We have made the code available in Zenodo for the version of the model with snow: (https://doi.org/10.5281/zenodo.5597877),
as well as for the version of the model without snow: (https://doi.org/10.5281/zenodo.5598076).

***Paragraph 2.2*** Along with calibration I strongly recommend to describe a validation procedure of the model.

We agree with the reviewer and appreciate this suggestion. As stated in the general comments, we have conducted several calibration-validation experiments following the recommendation of the reviewer. We explain the procedure in a new subsection 2.6 "Validation of the snow model" at the end of the Materials and Methods section (page 9, lines 273-280). The results of this validation are presented in the new subsection 3.1 "Model validation" added at the beginning of the Results section (page 10, lines 282-290), which

includes a new table (Table 3). Our results from the validation experiments show that the snow module is able to forecast snow reliably for the period not used for its calibration, and that the simulated snow is consistent using calibration models with different time spans.

**Paragraph 3.1** *I would recommend to calibrate and validate the new model on two independent time intervals.*
Following the reviewers recommendation, we split the simulation period in two separate time intervals and we performed three calibrations for the snow module for: (a) first half (1979-1996), second half (1997-2013), and for the full period (1979-2013). The three calibration models (the parameter sets obtained from the respective calibrations) were used to simulate snow for the full period of study and the simulated - snow properties were compared to snow observations. This is explained in the new subsection 2.6 "Validation of the snow model" (page 9, lines 273-280).

**Section 275.** *The authors wrote: "The values obtained for both sites were quite different". What does "quite different" mean? Are they statistically different?*
We agree with the reviewer that the expression "quite different" is not clear. We have eliminated this expression in favour of an objective statement indicating that we found the value of the resistance parameter to be almost twice as large in Tungsten than Caniapiscau (page 10, line 305).

**Section 275:** *This part "This potentially indicates that the physical factors that are neglected when assuming a constant ra (such as wind speed) are quite different between the two sites. The higher values at Tungsten indicates that snow sublimates at a slower rate, likely because the average wind speed during winter at this site is smaller than at Caniapiscau" is an interpretation which can be moved to discussion section*
We agree with the reviewer and this portion of text has been moved to the Discussion (page 13, lines 379-382).

**Sections 280-285** *can be moved to Discussion section.*
We agree with the reviewer that some of the contents of this paragraph do not belong to the Results section. We have relocated some of these contents (the description of the simulated water discharge and their sources, runoff and drainage) to the "Model evaluation and experiments" subsection in the Materials and Methods section (page 8, lines 244-248), while the interpretation on the effects of the snow module was repeated the Discussion (page 13, lines 383-394) and has therefore been removed from the Result section.

**Section 295.** *It seems to me the simulation GPP results with(without) snow are statistically the same (or even identical). Could the authors check it and indicate this in the MS?*
This is exactly the case, as we wanted to show that the snow module does not affect GPP. Therefore, the differences obtained in the calibration of TRC isotopes (which are weighted daily by GPP) derive only from the enrichment of xylem water due to snow. To make this point clearer, we have added this explanation and a t-test to show that both are statistically the same, in a new paragraph for the subsection "GPP calibration" in the Results (page 11, lines 325-329).

**Section 315** *can be moved to Discussion section.*

We agree with the reviewer and this interpretation has been moved to the Discussion in the revised version of the manuscript (page 14, lines 429-431).

**Section 330:** *Could the authors statistically check a difference between two simulations (Fig. 8C,*
We agree that a statistic will be more useful than the current qualitative statement to compare the curves produced by the two versions of the model. We have added a t-test (introduced in "Model evaluation and experiments", page 8, lines 241-243) to determine the difference between the two curves in each of the subfigures 8c and 8d (note that these have been renamed 8d and 8e respectively, due to the inclusion of a new 8c subfigure) for both sites. The obtained values are shown in these subfigures, and we have added a paragraph to explain the result of this test to subsection "GPP calibration" (page 11, lines 325-329).

**Figure 5:** *Where is the red curve on the Figure 5? Possibly, both versions (with/without snow) produce the same GPP results.*
Both curves are statistically identical (as determined by a t-test), thus the red curve is overlapped in the figure by the blue curve. We have added this explanation in the caption of Figure 5.

**Specific comments from Reviewer #2:**

**Page 4, lines 91-94:** *what type of mixing of soil water is incorporated in the model (for example between mobile and immobile soil water domains)?*
There is no distinction in the model between bound and mobile soil water, each soil layer (distributed vertically) acts as a single and homogeneous water reservoir and all water influx is considered to be mixed (completely and instantaneously) with the reservoir at every time step. The implementation of the isotopic model in MAIDEN (MAIDENiso, Danis et al., 2012) precedes the studies of McDonnell (2014), Evaristo et al. (2015) and Geris et al. (2015) on the two water worlds hypothesis (the separation of soil water into two reservoirs of bound and mobile soil waters with little connectivity between them, and preferential plant water uptake of bound soil water sources), and did not take into account the studies of Brooks et al. (2009) on the ecohydrologic separation of water between trees and streams. On the other hand, some studies contradict the assumptions for the two water worlds hypothesis (Vargas et al., 2017) or call for a satisfactory mechanism to describe persistent hydrological separation and more controlled hypothesis testing (Radolinski et al., 2021). Therefore, we do not think incorporating separate water domains in MAIDENiso is appropriate for this version of the model.

**Page 8, line 226 and 229:** *the GPP data from stations seems quite far from the study sites (1023 km and 650 km), with the assumption that obtained parameters were similar at the studied sites. Could you please elaborate a bit more why this assumption is (probably) valid?*
We agree with the reviewer that this assumption requires a bit more of justification than it currently has, and we have added an explanation in the revised version of the manuscript (page 8, lines 224-226). The GPP stations are indeed far from the study sites, but we chose these eddy-covariance flux stations because they provided GPP measurements for the same tree species. The parameters used to calibrate GPP in the model are more related to species-specific phenological processes than to environmental conditions in a given site. We

therefore think it is a reasonable assumption to use the GPP parameters determined in other sites for the same species.

***Page 8, line 245:*** *besides NSE, the Kling-Gupta efficiency (KGE), that is increasingly used as an alternative metric in hydrology, could provide useful additional information.*
We agree with the reviewer, the Kling-Gupta Efficiency supplements some of the deficiencies of the NSE. The description of the KGE has been added to the "Model evaluation and experiments" subsection in the Materials and methods (page 9, lines 256-262). We have used it in addition to NSE to determine the skills of the model to predict snow (page 9, lines 283-290) and river discharge (page 11, lines 316-318).

***Page 12, paragraph 4.1:*** *for the Discussion, but also for the results section, a nice addition is to provide delta18O for the hydrological outputs of the model. Many studies have reported that groundwater or streamwater does not show an evaporation fractionation signal compared to (top) soil or trees. Adding this to the analysis would make this study even more attractive to the more hydrological oriented readers of this journal.*
We thank the reviewer for this suggestion. We have added the isotopic composition of discharge water to Figure 8 (new panel c) and commented it in the Results section (page 12, lines 354-356) and the Discussion (page 14, lines 414-419). The model only produces discharge as an addition of runoff and the drainage water from the top layers, therefore it lacks any further complexity like the interaction with deeper groundwaters. Because of this the isotopic signal of the discharge water follows the signal of the xylem water. Despite this, we hope these outputs will be useful for the audience specialized in hydrology.

***Page 12, paragraph 4.4:*** *The current study could also be extended to sites that are snow dominated in the winter, but do have a Mediterranean climate, where trees are more dependent on water derived from snow melt, as this was not the case for the study sites as part of this study.*
We agree with the reviewer. Now that we have a working snow module, one of the means to extend this study would be to apply the new snow module to a network of moisture-dominated sites that also experience dry summers (South Western USA, Mediterranean sites), providing that we have sufficient local information on target variables (weather, snow, GPP, tree ring width and cellulose isotopes). These sites will be considered for future studies, and we have indicated it as such in a new paragraph in the "Implications for future studies" subsection in the Discussion (pages 15-16, lines 481-487).

We again thank the reviewers for their comments, which have allowed us to improve our paper significantly. On behalf of all the authors,

Ignacio Hermoso de Mendoza Naval

References:

- Brooks, J.R., Barnard, H.R., Coulombe, R. and McDonnell, J.J., 2010. Ecohydrologic separation of water between trees and streams in a Mediterranean climate. *Nature Geoscience. 3: 100-104*, *3*, pp.100-104.

- Danis, P.A., Hatté, C., Misson, L. and Guiot, J., 2012. MAIDENiso: a multiproxy biophysical model of tree-ring width and oxygen and carbon isotopes. *Canadian Journal of Forest Research*, *42*(9), pp.1697-1713.
- Evaristo, J., Jasechko, S. and McDonnell, J.J., 2015. Global separation of plant transpiration from groundwater and streamflow. *Nature*, *525*(7567), pp.91-94.
- Field, R.D., Andreu-Hayles, L., D'arrigo, R.D., Oelkers, R., Luckman, B.H., Morimoto, D., Boucher, E., Gennaretti, F., Hermoso, I., Lavergne, A. and Levesque, M., 2021. Tree-ring cellulose δ18O records similar large-scale climate influences as precipitation δ18O in the Northwest Territories of Canada. *Climate Dynamics*, pp.1-18.
- Geris, J., Tetzlaff, D., McDonnell, J., Anderson, J., Paton, G. and Soulsby, C., 2015. Ecohydrological separation in wet, low energy northern environments? A preliminary assessment using different soil water extraction techniques. *Hydrological Processes*, *29*(25), pp.5139-5152.
- McDonnell, J.J., 2014. The two water worlds hypothesis: ecohydrological separation of water between streams and trees?. *Wiley Interdisciplinary Reviews: Water*, *1*(4), pp.323-329.
- Radolinski, J., Pangle, L., Klaus, J. and Stewart, R.D., 2021. Testing the 'two water worlds' hypothesis under variable preferential flow conditions. *Hydrological Processes*, *35*(6), p.e14252.
- Vargas, A.I., Schaffer, B., Yuhong, L. and Sternberg, L.D.S.L., 2017. Testing plant use of mobile vs immobile soil water sources using stable isotope experiments. *New Phytologist*, *215*(2), pp.582-594.